



# Amplification of South Asian haze by water vapour-aerosol interactions

Vijayakumar Sivadasan Nair[1], Filippo Giorgi[2], Usha Keshav Hasyagar[1]

[1]Space Physics Laboratory, Vikram Sarabhai Space Centre, Thiruvananthapuram, Kerala, India
[2]Earth System Physics, International Centre for Theoretical Physics, Trieste, Italy

*Correspondence to*: Vijayakumar S. Nair (vijayakumarsnair@gmail.com)

**Abstract.** Air pollution and wintertime fog over South Asia is a major concern due to its significant implications on air quality, visibility and health. Using a coupled regional climate model with on-line chemistry, we assess the contribution of the hygroscopic growth of aerosols (wet-dry) to the total aerosol optical depth and demonstrate that the increased surface cooling due to the hygroscopic effects of aerosols further increases the humidity in the boundary layer and thus enhances the

confinement of pollutants through aerosol-boundary layer interactions. This positive feedback mechanism plays an important role in the prevalence of wintertime fog and low air quality conditions over South Asia, where water vapor contributes more than half of the aerosol optical depth. The aerosol-boundary layer interactions lead to the moistening of the boundary layer and drying of the free-troposphere, which amplifies the long-term trend in relative humidity over the Indo-Gangetic Plain during winter. Hence, the aerosol-water vapor interaction plays a decisive role in the formation and maintenance of the

wintertime thick fog conditions over South Asia, which needs to be considered for planning mitigation strategies.

## 1 Introduction

South Asia experiences regularly severe air pollution events during the winter season. Widespread haze and fog over the northern parts of the Indian sub-continent, especially the Indo Gangetic Plain (IGP), are associated with anthropogenic activities and are noticeable even from space during the winter season (Ali et al., 2019; Gautam and Singh,

2018; Ghude et al., 2017). Poor air quality and visibility persisting throughout the winter period has been a major concern for more than 900 million people living in the IGP (Gautam and Singh, 2018; Gurjar et al., 2008; Lelieveld et al., 2015). Indeed, various studies have shown that the winter concentrations of fine particles (PM2.5) and gaseous pollutants exceed national/international air quality standards over most of the IGP region and are considered extremely hazardous for human health (Ali et al., 2019; Bharali et al., 2019; Ghude et al., 2017; Gurjar et al., 2008; Kumar et al., 2015a; Nair et al., 2007;

Rengarajan et al., 2011; Safai et al., 2008). Recent estimates of premature mortality related to air pollution over India is ~0.65 million per year, mostly found to occur in the IGP due to residential emissions (Lelieveld et al., 2015). Traffic disruptions and air quality alerts are quite frequent during the winter season over megacities such as Delhi, Kolkata, Karachi, Lahore and Dhaka (Ghude et al., 2017; Gurjar et al., 2008). During the last few decades, the intensity of fog and haze events have increased over the region by a factor of three (Ghude et al., 2017; Syed et al., 2012), which is in-line with the observed



increase in aerosol optical loading (2.3% per year from 1985 onwards) and surface dimming over South Asia reported from a network of radiometer observations (Babu et al., 2013).

The sources of primary emission of anthropogenic aerosols include residential, transport, industrial and agricultural sectors (Kumar et al., 2015b). The extensive burning of stubble in agricultural fields has been reported to be a major contributor to the formation of haze at the beginning of winter. Mineral dust transport from west Asia and Thar deserts also

deteriorates air quality over the region. In addition to the emission sources, low temperatures, shallow boundary layers and low wind speed conditions hinder the vertical and horizontal mixing (ventilation) of aerosols, which favours the accumulation of pollutants within the boundary layer (Bharali et al., 2019; Nair et al., 2007). Even though the aerosol loading within the boundary layer is very high over the IGP, measurements onboard aircraft show relatively clean free troposphere conditions. The vertical extent of this high aerosol loading is thus limited and mostly confined to the first 2 km

near the surface with an exponential decrease above the boundary layer (Babu et al., 2016). Mass concentrations of fine particles (size < 2.5 μm, PM2.5) near the surface measured by the Central Pollution Control Board (CPCB) clearly indicate the poor air quality conditions prevailing over the IGP. The seasonal mean $PM_{2.5}$ values estimated over urban centers such as Delhi, Kanpur, Lucknow and Patna are higher than 150 μg m$^{-3}$ and even higher values (> 300 μg m$^{-3}$) are reported during fog events (Bharali et al., 2019). High aerosol loadings over the IGP prevail during most of the winter period, with more than

60% of days being foggy (Ghude et al., 2017). The seasonal mean value of PM2.5 reported at Delhi is three times higher than the national standard of 60 μg m$^{-3}$ for satisfactory levels of ambient air quality and the aerosol optical depth (AOD) exceeds 1.0 quite frequently over most of the IGP region.

In contrast to the numerous studies on the effects of aerosols on the Indian summer monsoon rainfall characteristics (e.g. see references in Li et al., (2016)), the impact of wintertime high aerosol loadings on regional climate and haze/fog

conditions have been little explored, although some studies have focused on the implications of regional emission sources on air quality or aerosol loading over the region. For example, Nair et al., (2007) showed that the aerosol variability over the IGP has strong relations with the boundary layer variability, whereas Bharali et al., (2019) reported that the aerosol forcing strongly influences the boundary layer evolution and thus strengthening the accumulation of pollutants near the surface. Hence, boundary layer variability affects the aerosol loading and the aerosol forcing influences the evolution of the boundary

layer as reported by several studies over polluted urban centres (Bharali et al., 2019; Ding et al., 2016; Huang et al., 2014; Li et al., 2017; Tie et al., 2017). Similarly, several observational studies investigated the correlation between the meteorological parameters, especially temperature and humidity with aerosol loadings (Kumar et al., 2015a). However, these observational studies have inherent difficulties in separating the aerosol forcing on meteorological parameters from the effects of meteorology on the aerosol loading. Generally, low temperature and high humidity are observed over IGP during winter,

which is influenced by the western disturbances. Wintertime precipitation over northern India is associated with western disturbances originating from the Mediterranean region, which bring cold-moist air over the aerosol-laden IGP. In addition, irrigation, water bodies (rivers and lakes) and farming activities enhance evapotranspiration and relative humidity in the



lower troposphere. These high humidity conditions, together with high aerosol loadings, lead to extensive and frequent fog events over the region (Gautam et al., 2007; Ghude et al., 2017; Goswami and Sarkar, 2015; Syed et al., 2012).

65       The effect of aerosols on the regional climate depends on the aerosol-radiation interactions, which are influenced by the water affinity of particles and the ambient relative humidity. The optical properties of aerosol increase by more than 2 times at higher relative humidity (>80%) condition, which has a strong dependence on the relative dominance of organic and inorganic species (chemical composition) and size of the particle. However, the direct measurements of hygroscopic growth functions for physical (size) or optical (scattering coefficient) of aerosols are limited over the Indian landmass (Mandariya et

al., 2020). Even though several studies addressed the implications of aerosols on air quality, health, and regional and global climate, there are very limited efforts to understand the implications of aerosol hygroscopicity on regional air quality. Based on these considerations, in this study, we use a regional climate model (the RegCM4, Giorgi et al., (2012)) coupled to air chemistry and aerosol model to assess the contribution of the aerosol hygroscopic growth to the total aerosol optical depth over the IGP. Since the total radiative impacts of aerosols on surface temperature, cloud properties and precipitation through

various forcing pathways are more comprehensively explored by several studies (eg: Li et al., (2016)), we focus only on the effects of the increase in AOD due to relative humidity and its implications on regional climate. Further, we quantify the effects and feedbacks of the aerosol hygroscopic growth on the low air quality conditions over the region. The model configuration and experimental details are given in the next section.

## 2 Materials and methods

**2.1 Regional Climate Model (RegCM)**

       For our study, we use the regional climate model version 4 (RegCM4) coupled with atmospheric chemistry and aerosol module. RegCM4 is a limited area model with a hydrostatic dynamical core and sigma vertical pressure coordinates (Giorgi et al., 2012). The South Asia-CORDEX domain is used in this study (Figure 01) with 50 km horizontal grid spacing and 18 vertical levels. The meteorological initial and lateral boundary conditions for our simulations are provided by the

ERA-interim re-analysis (Dee et al., 2011) for the three winter seasons (December to February) starting from November 2014 to March 2017, where the first month of simulation (November) of each year is discarded from the analysis as model spin-up. Optimum interpolation weekly sea surface temperature data from the National Oceanic and Atmospheric Administration (NOAA) are used as lower boundary condition over the Ocean. The parameterization schemes used in the simulations are (1) boundary layer: UW PBL scheme, (2) Convection over land and ocean: Tiedtke scheme, (3) Radiative

transfer: CCM3 scheme, (4) cloud microphysics: SUBEX and (5) Land surface: BATS scheme. More details on the model configuration and physics are already discussed in earlier papers (Ajay et al., 2019; Giorgi et al., 2012; Usha et al., 2020). Note that in the validation and analysis of the model output we will always consider the average over the three simulated seasons unless otherwise specified.





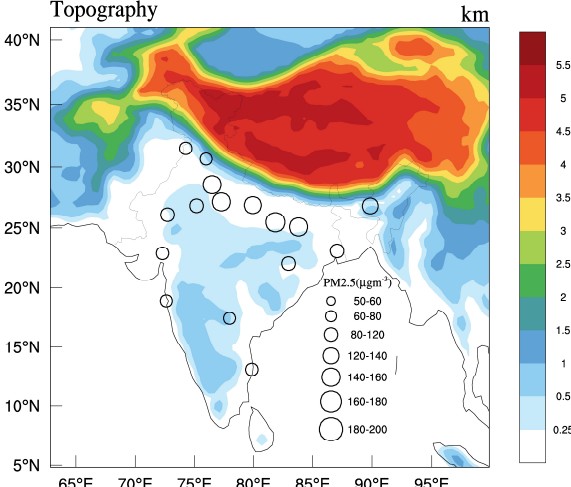

Figure 01: *Study domain centred over South Asia. Colormap shows the topography (km) and circles indicate the near-surface mass loading (µg m⁻³) of particles below 2.5 µm size range (PM2.5). PM2.5 data is taken from Central Pollution Control Board (CPCB), India.*

RegCM4 has an online chemistry module, which is extensively used for understanding aerosol-climate interactions (Nair et al., 2012; Shalaby et al., 2012; Solmon et al., 2015). The aerosol and trace gas emission fluxes are adopted from the 100 IIASA dataset, and chemical boundary conditions are from the global model MOZART (model of ozone and related chemical tracers). Gas-phase chemistry is based on the CBMZ (carbon-bond mechanism version Z) scheme and the ISORROPIA II is used for inorganic aerosols (Shalaby et al., 2012). The aerosol scheme includes sulphate, nitrate, ammonia, sea salt (2 size bins), mineral dust (4 size bins), black carbon (hydrophilic and hydrophobic) and organic carbon (hydrophilic and hydrophobic). The sources, sinks, atmospheric processes and transport of the aerosol species are detailed in Solmon et 105 al., (2006). The mass concentration of each aerosol species is converted into optical properties using a mass extinction cross-section, which depends on the ambient relative humidity and species-specific hygroscopic growth functions (Solmon et al., 2006). We use the growth function described by Kiehl et al., (2000) for sulphate, nitrate and ammonia aerosols. In this scheme, the growth function increases exponentially with humidity. For example, at 80% relative humidity, the particle mass extinction cross-section increases by a factor of 3 compared to its dry value. The model emits carbonaceous aerosols as 110 hydrophobic and the optical properties of this nascent carbonaceous aerosols are invariant with relative humidity. These particles changes from hydrophobic to hydrophilic due to ageing at a fixed time scale of 1.15 days (~27.6 hours, Solmon et al., (2006)). The hydrophilic part of the carbonaceous aerosols (black carbon and organic carbon) has a weak affinity to water compared to sulphate aerosols (Kasten, 1969). At 80% relative humidity, the growth factor is 1.37 and 1.49 for





hydrophilic black carbon and organic carbon, respectively. Mineral dust is considered as fully hydrophobic. The sea salt size

increases by 2 times at 80% relative humidity compared to dry conditions, but this is of little importance over the IGP during

winter. In general, most of the aerosols grow with relative humidity and this enhances the optical depth.

**2.2 Experiments**

RegCM has been widely used to investigate aerosol-hydroclimate interactions and the impacts of various forcings

on monsoon characteristics (e.g. Solmon et al., (2015) and Usha et al., (2020)). The model can be run with and without

climate feedbacks of aerosols. In the former case, aerosols affect the radiation balance and change the surface temperature

and meteorology through direct and indirect pathways. In the latter, the aerosol radiative forcing is calculated but it does not

have effects on the meteorology, surface temperature and atmospheric thermodynamics. In the present study, model

simulations with (ambient) and without (dry) hygroscopic growth of aerosols are named as experiments 1 and 2 respectively.

Experimental details are summarized in Table 1. In the case of dry aerosol simulations, we forced the aerosol hygroscopic

growth functions to be equal to 1 in the model code. The model runs without climate feedbacks (Exp1 and Exp2) are

considered as control runs for wet and dry aerosol cases. Since the effect of aerosol-radiation interactions on land and

meteorological parameters are switched off (no climate feedback), the difference between these two experiments provides

the contribution of hygroscopic growth to the aerosol properties ($\Delta AOD_{RH}$)

$$\Delta AOD_{RH} = AOD(Ambient_{nofeedback}) - AOD(Dry_{nofeedback}) \tag{1}$$

Experiments 3 and 4 are similar to Exp 1 and 2 but with climate feedbacks (Table 1). The difference between the Exp3 and 4

provides the net effect of the hygroscopic growth of aerosols on the regional climate, which includes (i) changes due to the

radiative forcing of the aerosol hygroscopic growth and (ii) its climate feedback. For example, the change in temperature (T)

due to the climate feedback of $\Delta AOD_{RH}$ is estimated as

$$\Delta T_{RH} = T(Ambient_{feedback}) - T(Dry_{feedback}) \tag{2}$$

The effect of total aerosol system on meteorological variables (for eg: Temperature) is estimated as

$$\Delta T_{aerosol} = T(Ambient_{feedback}) - T(Ambeint_{nofeedback}) \tag{3}$$

The change in AOD solely due to climate feedback of total aerosol system (dry + hygroscopic growth) is estimated as

$$\Delta AOD_{feedback} = AOD(Ambient_{feedback}) - AOD(Ambient_{nofeedback}) \tag{4}$$

Hence the AOD(Ambient$_{feedback}$) is the sum of dry AOD, its hygroscopic growth and climate feedback

$$AOD(Ambient_{feedback}) = AOD(Dry_{nofeedback}) + \Delta AOD_{RH} + \Delta AOD_{feedback} \tag{5}$$

Table 1: Experimental details.

| Exp. Name | Hygroscopic growth | Climate feedback |
|---|---|---|
| Exp1 : Ambient$_{nofeedback}$ | Yes | No |
| Exp2: Dry$_{nofeedback}$ | No | No |
| Exp3: Ambient$_{feedback}$ | Yes | Yes |



| Exp4: Dry_feedback | No | Yes |
|---|---|---|

**2.3 Model Validation**

RegCM simulates reasonably well the meteorology of South Asia (Ajay et al., 2019; Giorgi et al., 2012; Nair et al., 2012; Usha et al., 2020). In general, the model captures the basic features of seasonal precipitation (monsoon), though it is
sensitive to the convection schemes used and the extent of the ocean-atmosphere coupling (Ajay et al., 2019). Studies on the validation of chemical constituents simulated by the model are rather limited over South Asia (Nair et al., 2012; Solmon et al., 2015; Usha et al., 2020). A qualitative inter-comparison of the chemical composition of aerosols simulated by RegCM with in-situ measurements at distinct locations over the Indian region is shown in Figure 2. The measurements of column aerosol optical depth are taken from AERONET (Aerosol Robotic Network) radiometer observations at Jaipur, Kanpur,
Gandhi College and Delhi. The black carbon measurements were carried out using Aethalometers installed under the ARFI (Aerosol Radiative Forcing over India) project. Since the measurements of aerosol composition are sparse over the Indian region, we have taken values reported from earlier studies for the winter season, and therefore the organic carbon and sulphate measurements are for different years than those of the model simulations (Ali et al., 2019; Aswini et al., 2019; George et al., 2008; Ram et al., 2010, 2012; Rengarajan et al., 2011; Safai et al., 2008; Satsangi et al., 2012). Despite these
shortcomings, compared to earlier studies, the present model configuration simulates black carbon mass loading and AOD which are closer to the observed values (Nair et al., 2012). Though RegCM has a simple scheme for organic aerosols, the organic carbon mass concentration broadly matches the observed seasonal mean values. In fact, most climate and chemical transport models fail to capture the high aerosol loading over this region, a problem which has been mostly attributed to the parameterization of stable boundary layer conditions (Nair et al., 2012), unaccounted emissions (eg: Nair et al., (2012),
references are therein) and biases in simulating relative humidity and precipitation (Chatani and Sharma, 2018; Feng et al., 2016). In the present study, AOD and the near-surface mass concentrations of black carbon, organic carbon, sulphate and dust simulated by the model are in the range of values reported by several studies over the region. One of the main challenges in simulating the AOD over the region is the accurate simulation of relative humidity, which affects hygroscopic growth (Chatani and Sharma, 2018). The mean relative humidity estimated from several re-analysis datasets and
measurements in different locations of the IGP varies from 60 to 80% (Chatani and Sharma, 2018; Gautam et al., 2007; Ghude et al., 2017; Goswami and Sarkar, 2015). A good agreement is found between measured and simulated (with aerosol feedback) relative humidity over the IGP (Fig. 2e), which is important for the estimation of aerosol optical properties from simulated speciated aerosol mass concentrations. During winter, aerosols are confined within the boundary layer, and changes in the boundary layer height play a major role in the dilution (ventilation) of aerosols over the IGP (Bharali et al.,
2019; Nair et al., 2012). Shallow and stable boundary layers prevail during the winter season and a comparison with IGRA (Integrated Global Radiosonde Archive) radiosonde data indicates that the UW scheme is able to simulate the boundary layer height over the region reasonably well (Fig. 2f).

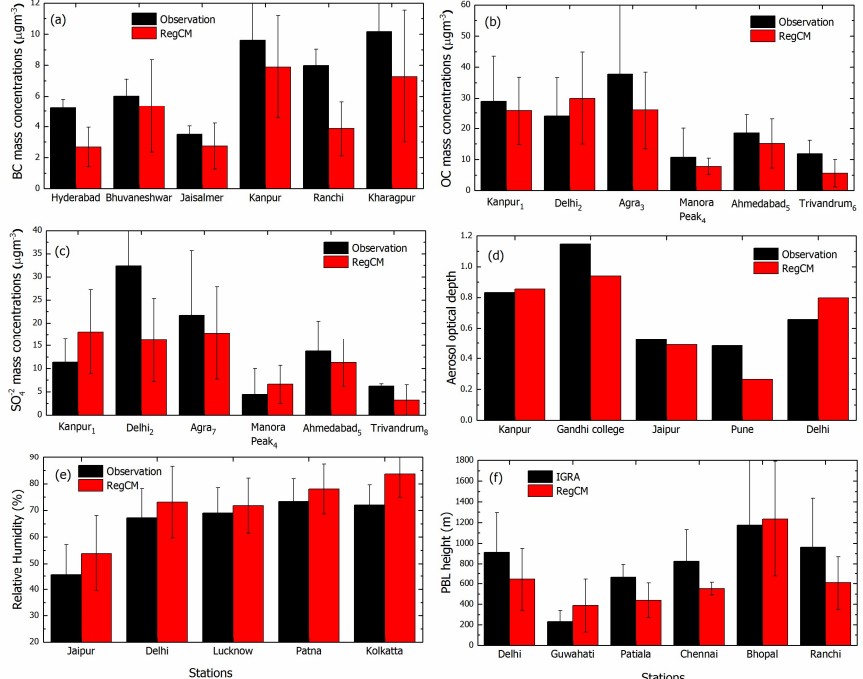

Figure 2: *Validation of RegCM simulated aerosol parameters ((a) black carbon (BC), (b) organic carbon (OC), (c) sulphate, (d) AOD), (e) relative humidity and (f) boundary layer height with measurements over different locations in India. The measurements of OC and sulphate mass loading are taken from the earlier measurements (Kanpur[1] - Ram et al., (2012); Delhi[2] - Ali et al., (2019); Agra[3] - Satsangi et al., (2012); Manora Peak[4] - Ram et al., (2010); Ahmedabad[5] - Rengarajan et al., (2011); Trivandrum[6] - Aswini et al., (2019); Agra[7] - Safai et al., (2008); Trivandrum[8] - George et al., (2008)) and these values represent the winter season but for different years than those of the model simulations.*

## 3 Results and discussions

### 3.1 Hygroscopic growth of aerosols

The particulate mass loadings (PM2.5) measured over most cities in the IGP during winter are well above the air quality standards, as shown in Figure 1. The total column AOD at ambient humidity simulated by RegCM4 with climate feedback shows very high values over the IGP, mostly consisting of boundary layer aerosols. The comparison of measured and simulated AOD for dry and ambient humidity conditions (Fig. 3) over the distinct environments of IGP indicates an almost two-fold increase in AOD due to the hygroscopic growth of aerosols and its climate feedback. The mean observed



(ambient, dry) AOD for Jaipur, Kanpur, Gandhi College and Pune are 0.53 (0.51, 0.26), 0.83 (0.81, 0.34), 1.15 (0.92, 0.36) and 0.48 (0.32, 0.15) respectively. In general, both the magnitude and large day-to-day variability in AOD (Fig. 3) are captured by the model when the hygroscopic growth ($\Delta AOD_{RH}$) and climate feedback ($\Delta AOD_{feedback}$) are included in the

simulation. Except for dust and freshly emitted black carbon, most of the aerosol species have an affinity towards water vapour, with the hygroscopic growth of inorganic aerosols such as sulphate, ammonia, nitrate and sea salt being more pronounced compared to organic carbon (Solmon et al., 2006). Due to ageing a fraction of hydrophobic black carbon also becomes slightly hydrophilic (Solmon et al., 2006) and contribute, though small, to the AOD at ambient humidity. Observations close to the Thar Desert (Fig. 1a, Jaipur) do not show much hygroscopic growth (43%) because of the

dominance of dust aerosols and relatively low humidity conditions prevailing there.

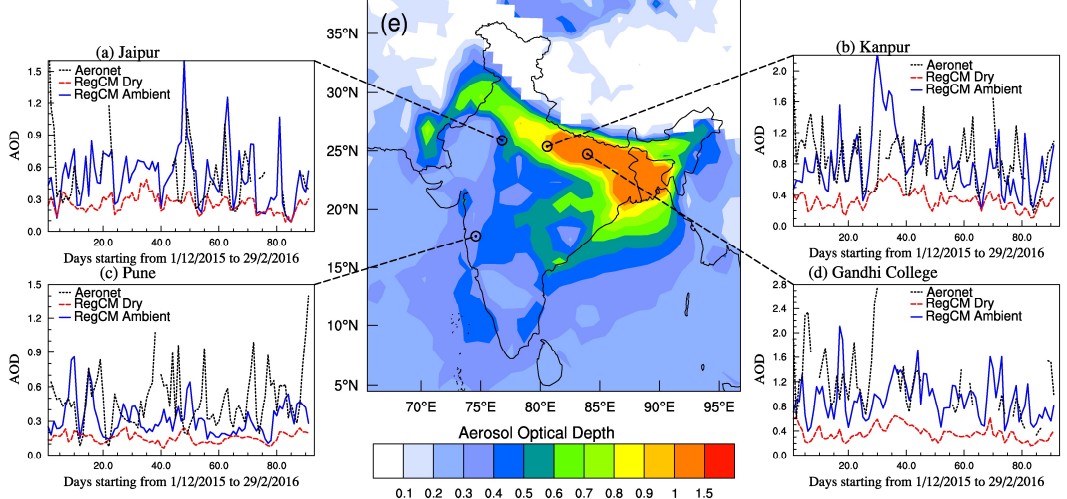

Figure 3: *Wintertime AOD measured using Aeronet radiometers and simulated with RegCM4 for ambient and dry humidity conditions with climate feedback at (a) Jaipur, (b) Kanpur, (c) Pune and (d) Gandhi College. Stations (b) and (d) are located into the IGP while (a) and (c) are located outside the IGP. One season (December 2015 - February 2016) data is*

*shown here for inter-comparison; same is the case for other years. (e) Spatial distribution of satellite (MODIS) retrieved aerosol optical depth at 550 nm.*

Our study shows that the large day-to-day and spatial variability in AOD over the IGP is mostly contributed by relative humidity (Fig. 3), with lower contributions from changes in source characteristics and synoptic-scale circulation. The simulated AOD at ambient relative humidity matches well with the measured AOD whereas the dry AOD is

significantly lower and less variable in time. The standard deviation of measured and simulated AOD at ambient humidity



conditions is nearly half of the mean AODs (coefficient of variation of 50%) (Fig. 3), whereas, for dry aerosols, the standard deviation of the AOD is low (0.05 to 0.1), with a coefficient of variance ranging between 20 to 30%. Thus, since the emission fluxes of anthropogenic aerosols are seasonally invariant, most of the observed variability of AOD is due to the variability in relative humidity. The AOD due to dry aerosols has 10 to 15 % less variability compared to that of the total

AOD. Over these stations, the $AOD(Ambient_{nofeedback})$ contribute almost 80 to 90 % of the $AOD(Ambient_{feedback})$.

The AOD due to hygroscopic growth ($\Delta AOD_{RH}$) estimated using Eqn. 1 is shown in Fig. 4a. The $\Delta AOD_{RH}$ values are high over IGP, especially the eastern IGP and Bangladesh ($\Delta AOD_{RH} > 0.3$). The ratio of $AOD(Dry_{nofeedback})$ to $AOD(Ambient_{nofeedback})$ shown as contours in Figure 4a indicates that dry aerosols contribute 50 to 60% of the AOD over most of the Indian sub-continent (Fig. 4a), except over western India, where the contribution of hydrophobic dust to the total

AOD is high even during winter (Babu et al., 2016). The contribution of the aerosol hygroscopic growth is more than 50% of the total AOD over the northern Indian Ocean, which is attributed to the dominance of hydrophilic aerosols (sea salt) and high relative humidity. Wintertime aerosols over central IGP shows an increase in particle diameter by 1.34±0.07 times as that of the dry particle at 85% relative humidity (Mandariya et al., 2020). In the case of anthropogenic aerosols, aerosol scattering coefficient increases more than twice when the particle diameter increase by 1.34 times of its dry diameter

(McInnes et al., 1998). However, the mean dry AOD increases almost 1.7 times due to relative humidity over IGP. Since organic aerosols dominate the aerosol system over IGP (Ram et al., 2012), there could be a substantial decrease in the hygroscopic growth of particles as discussed by several studies. Mandariya et al., (2020) have reported that 150 nm particles are more hygroscopic than 100 nm particles due to the higher organic fraction present in the 100 nm particle mass concentration. In addition, although most of the observational studies consider 40% as the reference humidity level (dry) for

estimating the hygroscopic growth, the model estimates the dry AOD for zero relative humidity. Titos et al., (2016) reported the assumption of no growth at 40% humidity contribute to higher uncertainty in the estimates of measurement-based hygroscopic growth. The increase in AOD due to hygroscopic growth ($\Delta AOD_{RH}$ = AOD(ambient) – AOD(dry)) is ~0.23 over the IGP (Fig. 4a), which is higher than the global mean AOD (0.155±0.018) reported by Watson-Parris et al., (2020) using various satellite observations.

In general, the mean AOD over the IGP stations (Kanpur and Gandhi College) is ~0.7, indicating significant surface dimming of solar radiation (Fig. 4b). The increase in AOD due to hygroscopic growth alone reduces the surface solar flux by 10 to 20 Wm$^{-2}$ over the IGP. The surface dimming due to aerosol-radiation interaction results in the reduction of surface temperature, heat fluxes (sensible and latent) and boundary layer height (Bharali et al., 2019; Ding et al., 2016; Li et al., 2017). As a result, aerosol forcing leads to the accumulation of pollutants in the boundary layer (as discussed later).

Since water vapor contributes significantly to the total AOD, most of the implications of aerosols on regional climate, visibility and air quality are strongly associated with meteorological conditions.


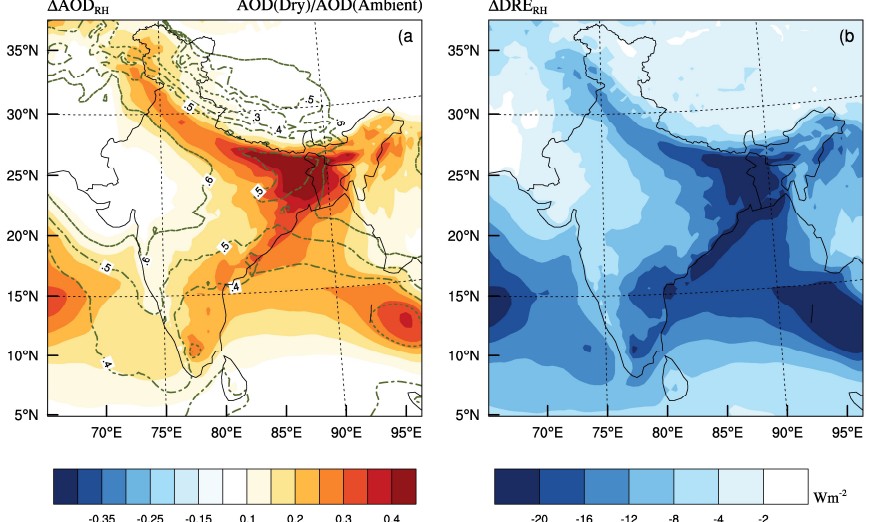

Figure 4: *(a) Aerosol optical depth (AOD) simulated using RegCM for dry and ambient humidity conditions. Colour map indicates AOD(ambient) – AOD(dry) for without climate feedback and line contours show AOD(dry)/AOD(ambient) (b) Difference in aerosol direct radiative effects (DRE) at the surface for dry and ambient humidity conditions (without climate feedbacks).*

### 3.2 Effects on regional meteorology

The effects of the increase in AOD solely due to the hygroscopic growth of particles ($\Delta AOD_{RH}$) on surface temperature and relative humidity (Eqn. 2), illustrated in Figure 5 (top panel), showing a surface cooling of about 0.5°C and a 3% increase in relative humidity over the IGP during winter. This is mostly due to the direct solar dimming at the surface and to a cloud cover change induced by the $\Delta AOD_{RH}$. The total effect of aerosols on temperature and relative humidity (Eqn. 3) is shown in Figures 5c & 5d. Nearly 40 to 50% of the aerosol-induced surface cooling is contributed by the water vapor through the hygroscopic growth of particles. These effects are more prominent over the IGP and central India, whereas high $\Delta AOD_{RH}$ and surface dimming was seen over eastern IGP (Fig. 4). Although AOD and $\Delta AOD_{RH}$ are high, the change in relative humidity due to hygroscopic growth of aerosols is negligible over Bangladesh and Eastern IGP. Hence the strength of land-atmosphere interactions through the exchange of heat and moisture fluxes plays a major role in deciding the aerosol induced dimming effects on meteorology (Bharali et al., 2019; Li et al., 2017). At the surface, an observed decrease in evapotranspiration (and associated rainfall) due to solar dimming further confirms a slowing down of the hydrological cycle induced by the aerosol hygroscopic growth (Liepert et al., 2004; Ramanathan, 2001). In other words, water vapour itself reduces the evaporation through dimming associated with the aerosol hygroscopic growth.



The mean vertical profiles of change in temperature and relative humidity over the IGP due to total aerosol and hygroscopic effects are shown in Figure 6. Significant cooling and associated moistening are noticeable in the boundary layer followed by a weak drying in the lower free troposphere (1 to 3 km) and a negligible influence above (>3 km). Based on extensive measurements from Delhi during wintertime fog campaign, Ghude et al., (2017) have reported more than 70%

humidity within the boundary layer and dry condition (<20%) above 3 km. A similar pattern is observed for specific humidity with the highest increase of 0.6 g kg$^{-1}$ at the surface. This figure further confirms that the hygroscopic growth is the single largest contributor to the total aerosol effects over the IGP during winter.

The surface cooling, along with the weak entrainment of dry airmass from the free troposphere, leads to an increase in relative and specific humidity at the surface (Fig. 5a). Based on the extensive measurements of aerosol and humidity

profiles, Feng et al., (2016) reported up to 30% underestimation of relative humidity within the boundary layer and 20% overestimation in the free troposphere by WRF-Chem model over a north Indian station (Nainital) during the winter season. These authors attributed this dry bias in the boundary layer humidity as one of the major reasons for the underestimation of AOD by climate models over the region. Most re-analysis datasets (NCEP and ECMWF) have a negative bias (underestimate) in simulating relative humidity over this region, especially during winter (Chatani and Sharma, 2018; Feng

et al., 2016), which significantly affects the simulation of aerosol optical depth, aerosol radiative forcing and fog prediction. The present study shows that the negative bias (underestimation) of relative humidity could be reduced, especially at a high relative humidity (RH > 70%), by including the climate feedback of ambient aerosols (Exp 4). In contrast to boundary layer moistening, drying in the lower free troposphere (1 to 3 km) by 1 to 2% was attributed to the warming of the top of the boundary layer, which decreases the humidity and suppresses the mixing of airmass between the free troposphere and

boundary layer. Compared to dry condition, the warming of the top of the boundary layer occurs at a lower altitude for ambient aerosol forcing.

We analysed re-analysis data and in-situ observations for the long-term trend in wintertime relative humidity within the boundary layer and free-troposphere over the IGP during the last four decades (Fig. 7). In contrast to the global scenario, relative humidity is increasing over the Indian region at the rate of 1% per decade, together with a concurrent increase in

aerosol loadings (Babu et al., 2013) and the number of foggy days (Ghude et al., 2017; Syed et al., 2012). Based on the above discussions (Fig. 6), it can be argued that the aerosol forcing contributes significantly to the observed increasing trend of wintertime moistening of the boundary layer and enhanced occurrence of fog events over the IGP. The aerosol radiative forcing increases the stability of the atmosphere and decreases the transport of moisture from the boundary layer to the free troposphere, which favours the drying trend in the lower free troposphere as shown in Figure 7b. Earlier studies also

attributed the humidification of the boundary layer and drying of the free troposphere during winter to the aerosol forcing (Li et al., 2017; Tie et al., 2017). Our analysis shows that aerosol feedback processes significantly increase near-surface relative humidity due to weak turbulent diffusivity in the stratified boundary layer (Bharali et al., 2019; Li et al., 2017). The increase in relative humidity, when surface temperature increases, is due to the increase in water vapour content in the atmosphere as seen in the specific humidity trend reported by Mukhopadhyay et al., (2017). In contrast to the negative trend in relative



humidity at lower free troposphere during winter (Fig. 7b), Mukhopadhyay et al., (2017) show a positive trend for annual
mean humidity, which is largely dominated by the positive trend during summer. Global analyses have shown that most
continental regions exhibit a long term warming trend and decrease in relative humidity (Dai, 2006), which has strong
implications for the land-ocean warming contrast (Hodnebrog et al., 2019). The long-term trend in the moistening of the
boundary layer and drying of the free troposphere over the Indian region during winter is further amplified by the aerosol

radiative forcing, primarily through the hygroscopic growth. However, the masking effects of surface warming by aerosols
and the contribution of aerosols to the increasing trend in humidity is yet to be quantified.

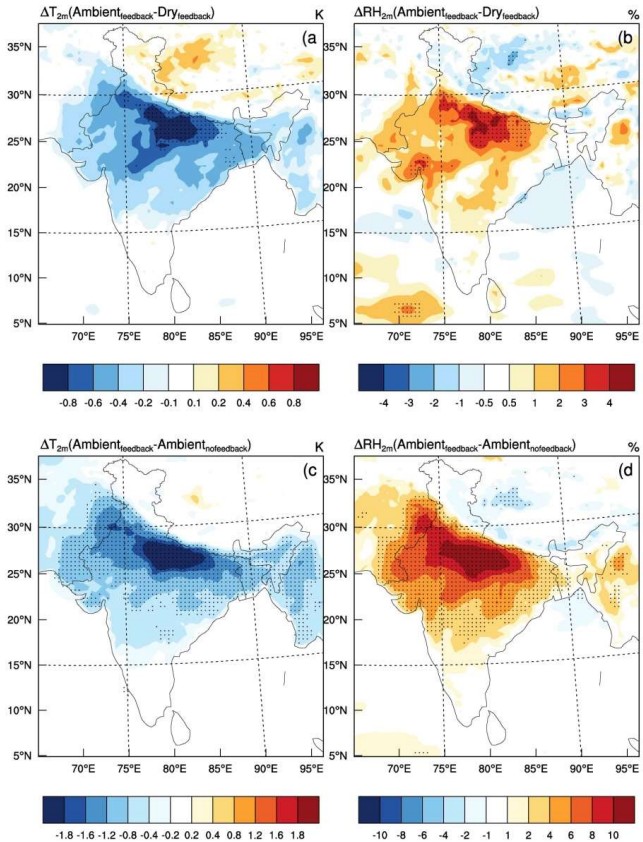

Figure 5: *Change in (a) 2m air temperature and (b) relative humidity solely due to the hygroscopic growth of aerosols*
*(ambient-dry). Change in temperature and relative humidity (RH) due to the total effect of aerosols (dry + hygroscopic*
*growth) is shown (c) and (d). Dots indicate statistical significance of Student's t-test at a 90% level.*



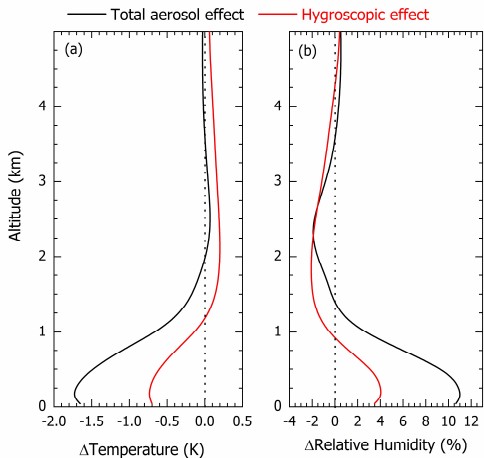

Figure 6: *Vertical profiles of change in relative humidity and temperature due to the total aerosol effect (Ambient$_{feedback}$ - Ambient$_{nofeedback}$) (black line) and only due to hygroscopic growth of aerosols (Ambient$_{feedback}$ - dry$_{feedback}$).*

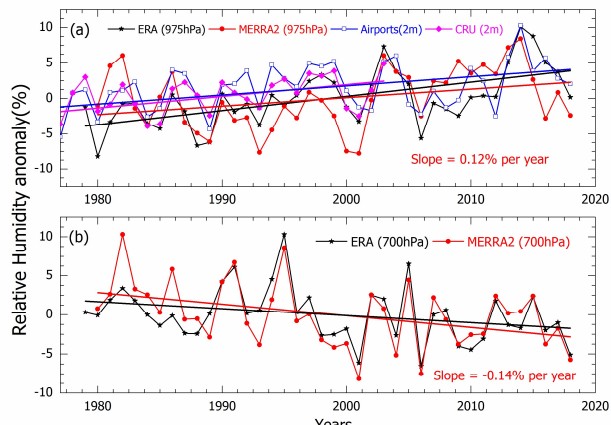

Figure 7: *(a) Relative humidity in the boundary layer (975hPa) from re-analysis (ERA, MERRA and CRU) and in-situ observations (b) relative humidity in the free troposphere (700 hPa) taken from re-analysis data.*

### 3.3 Implications on air quality

Regional mean AOD increase of ~0.23 is contributed by the hygroscopic growth of aerosols as shown in Figure 3. This hygroscopically grown aerosol system further increases the AOD through climate feedbacks ($\Delta AOD_{feedback}$), whose



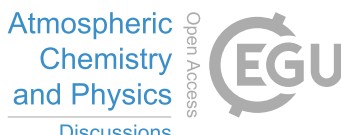

contribution can be estimated using Eqn. 4 and is shown in Figure 8a. Overall, hygroscopic growth ($\Delta AOD_{RH} \sim 0.23$) and associated climate feedback ($\Delta AOD_{feedback} \sim 0.17$) contribute significantly to the total AOD (0.72) over the IGP. This positive feedback is stronger over the central IGP than the eastern IGP, though eastern IGP has higher AOD and relative humidity. The moisture content in the atmosphere increases the AOD directly by its hygroscopic effect (Fig. 3) while high AOD increases the surface relative humidity through its radiative and climate feedbacks (Fig. 8a). Even though the spatial

patterns of change in AOD and relative humidity due to aerosol feedbacks are slightly different, the anomalies in AOD and relative humidity show a significant correlation over most of the Indian sub-continent, especially over the IGP (Fig. 8b).

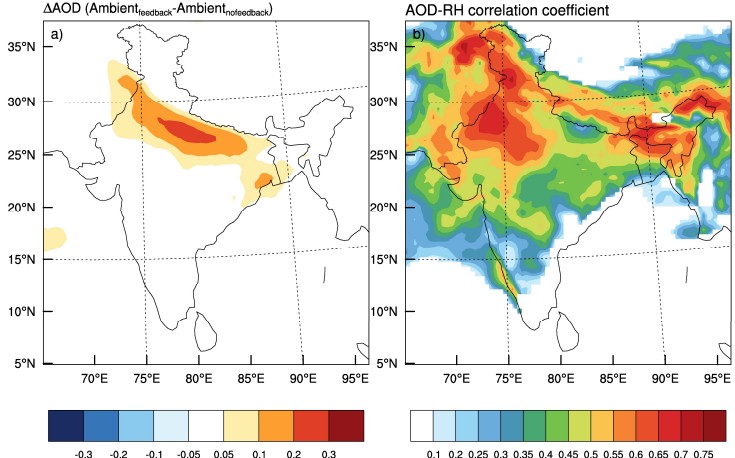

Figure 8: *Change in AOD due to the climate feedbacks of the hygroscopic growth of aerosols ((Ambient_feedback-Dry_feedback)-(Ambient_nofeedback-Dry_nofeedback)). Correlation coefficient of change in AOD and relative humidity due to the climate feedback*

*of hygroscopic growth of aerosols.*

The frequency of occurrence of AOD and PM2.5 for dry and ambient humidity conditions with climate feedback over the IGP is shown in Figure 9. The inclusion of climate feedback and hygroscopic aerosol growth ($\Delta AOD_{RH}$) produces an increase in AOD along the IGP (Fig. 5). The narrow frequency distribution for dry AOD changes to a broad pattern for AOD at ambient humidity, which implies an increase in the number of days with hazy skies and significant dimming at

surface. There is a substantial increase in the number of days having high AOD for ambient humidity condition. The frequency distribution of the ratio of AOD for ambient and dry humidity with and without climate feedback is shown in Figure 9b. The narrow distribution for no-feedback simulation with a mean ratio of 1.72 changes to a broad distribution having a mean increase of dry AOD by 2.3 times at ambient humidity with climate feedbacks. Due to the positive feedback as mentioned above, AOD at ambient humidity increases more than 3 times of its dry AOD, depending on the strength of the

land-atmosphere and aerosol-boundary layer interaction. The magnitude of the climate feedback due to the total aerosol





system ($\Delta AOD_{feedback}$) is comparable to $\Delta AOD_{RH}$ over IGP. This clearly indicates that not only the hygroscopic growth of aerosols, climate feedback is also important over IGP. The implications of aerosol on regional meteorology and effects this change in meteorology on aerosol loading need to be investigated further to better understand and predict the fog events over IGP (Bharali et al., 2019). Although the dominance of organics in the aerosol mass loading over IGP is expected to decrease

the hygroscopic growth of pure inorganic aerosols (Mandariya et al., 2020), the climate feedback processes (Eqn. 3) increases the AOD by moistening the boundary layer ($\Delta RH_{aerosol}$). The frequency of occurrence of high values of PM2.5 mass concentration for ambient humidity shows deterioration of air quality due to aerosol-radiation interaction. Since model estimate PM2.5 mass concentration for dry condition (without hygroscopic growth), PM2.5 has less difference between the dry and ambient humidity compared to that of AOD. Hence this analysis shows that aerosol forcing itself increases the

aerosol loading through aerosol-boundary layer interaction.

     In fact, generally calm winds (<2 ms$^{-1}$), cold temperature, shallow boundary layer and slow descending motion prevail over the IGP during winter which favours the accumulation of aerosols within the boundary layer (Ghude et al., 2017; Kumar et al., 2015a; Nair et al., 2007). In addition, the aerosol effects further weaken the horizontal and vertical circulations and decrease the boundary layer height, which results in more accumulation of aerosols and thus poor air quality

(Bharali et al., 2019; Ding et al., 2016). Although wintertime aerosols are mostly confined within the boundary layer, near-surface PM2.5 and columnar aerosol loading show poor correlation over the IGP as reported by Kumar et al., (2015a). The present study also showed poor association with ambient AOD and PM2.5, but the high correlation is observed between the dry AOD and PM2.5. Briefly for deriving the surface mass concentration from spaceborne sensors, which is very important for the air quality studies, the contribution of humidity to the PM2.5 and AOD measurements should be addressed carefully.

By and large, the offline filter-based PM2.5 measurements have issues associated with sampling aerosols at ambient humidity conditions. The mass concentration is dominated by the contribution from the coarse mode aerosols and AOD by fine/accumulation mode aerosols. This also points to the need for dedicated field experiments focussing on the size segregated hygroscopic growth functions of physical and optical properties of aerosols over the IGP (Mandariya et al., 2020).

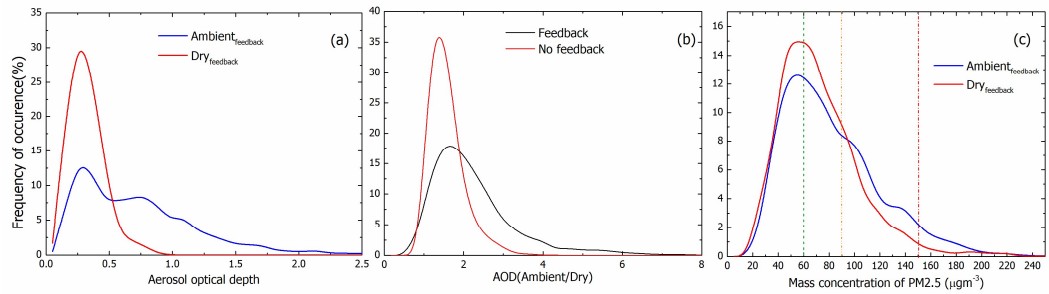




Figure 9: *Frequency of occurrence of (a) AOD and (c) PM2.5 for dry and ambient relative humidity conditions with climate feedback over the Indo-Gangetic Plain (IGP). (b) The frequency distribution of the ratio of AOD for ambient and dry humidity with and without climate feedback. Dotted vertical lines on PM2.5 frequency of occurrence shows the breakpoint concentration for air quality index category for satisfactory (60 µg m⁻³), moderately polluted (90 µg m⁻³) and poor (150 µg*

*m⁻³) conditions as proposed by Central Pollution Control Board (CPCB), India.*

As mentioned above, the AOD increase via hygroscopic growth of aerosols under high relative humidity conditions further decreases the incoming solar radiation at the surface, which results in enhanced surface cooling (compared to dry aerosols) and a decrease of the exchanges between the boundary layer and free troposphere. Aerosol-radiation interactions lead to a significant increase in cloud cover over the IGP region (2 to 8%), which is further increased by the hygroscopic

effect of aerosols. The increase in cloud cover leads to more sulphate formation and thus again influence the radiative balance. Hence hygroscopic effects of aerosols significantly strengthen the observed aerosol-boundary layer interactions over the region (Bharali et al., 2019). Recent studies have shown that the high aerosol concentrations at the surface over the urban centres are strongly related to positive feedback processes associated with the boundary layer and water vapour (Ding et al., 2016; Huang et al., 2014; Li et al., 2017; Tie et al., 2017). The surface solar dimming and atmospheric warming due to

black carbon in the upper boundary layer decrease the height of the mixed layer (increases stratification) and increase the accumulation of aerosols within the boundary layer (Bharali et al., 2019; Ding et al., 2016; Nair et al., 2007). The aerosol induced increase in relative humidity (1-10 %) over most land regions (Fig. 5), especially the central IGP, further increases the AOD as shown above. Tie et al., (2017) have reported that a decrease in the dispersion of water vapour leads to a self-amplifying feedback mechanism through which an increase in relative humidity further increases the AOD due to

hygroscopic growth ($\Delta AOD_{RH}$). All these pathways further increase moisture in the boundary layer and, consequently, the AOD (Tie et al., 2017). These feedback processes significantly deteriorate air quality conditions during winter (Li et al., 2017).

## 4 Conclusions

In this work, the regional climate model (RegCM4) coupled with atmospheric chemistry is used to investigate the

contribution of the hygroscopic growth of aerosols to the total aerosol optical depth and its climate feedback over the Indian sub-continent. Our analysis shows that the aerosol hygroscopic growth can contribute up to 40% of the total AOD and that feedback processes significantly increase near-surface relative humidity and decrease lower free troposphere humidity. This might strengthen the long-term trend in boundary layer moistening and free tropospheric drying over the IGP during winter. We also show that the inclusion of the climate feedback due to hygroscopic growth of aerosols ($\Delta AOD_{RH}$) produces an

increase in AOD and PM2.5 along the IGP. This positive feedback is stronger over the central IGP than the eastern IGP. The moisture content in the atmosphere not only increases the AOD directly by its hygroscopic effect but also indirectly through its radiative and climate feedbacks. Briefly, aerosol forcing due to dry aerosols and its hygroscopic growth at high relative





humidity conditions increase the relative humidity in the boundary layer, which further increase the AOD through a positive feedback. Our analysis demonstrates that the aerosol-moisture interaction is the most significant contributor favouring and

strengthening the high aerosol conditions (poor air quality) prevailing over IGP during winter.

Our study also highlights the need to increase understanding of aerosol/climate/air quality interactions over the India sub-continent: i) including the simulation of hygroscopic growth and related feedbacks in climate/chemistry models; ii) direct measurements of hygroscopic growth functions of aerosols, which are rather limited over the region (Mandariya et al., 2020); and iii) measurements and model description of the effect of ageing and mixing state on the water affinity of

hydrophobic aerosols. The model limitations specific to this study are worth noting here. The RegCM has a simple organic aerosol module and a single growth function for the hundreds of organic species present in the atmosphere, which are characterized by a wide range of affinity towards water vapour. The model does not include the effects of organic aerosols on the water affinity of inorganic aerosols. All these aspects of model development are underway in the next version of the RegCM modelling system.

Notwithstanding these limitations, our study clearly shows that understanding the interactions of natural factors (moisture fluxes and relative humidity) with the anthropogenic aerosols (organic and inorganic) is essential for predicting fog and haze events and devising pollution mitigation strategies over the IGP. In this regard, the aerosol-water vapour interaction is a unique example of the amplification of anthropogenic forcing (aerosols) by natural agents (water vapour) leading to significant changes in regional climate and air quality. To date low air quality and visibility events over the IGP

have been considered essentially as a problem of emission sources and transport of particles, however, our study highlights the important, and in fact sometimes dominant, the contribution of atmospheric water vapour to these events, and thus the need to consider this natural factor in air quality assessments and related policymaking.

**Acknowledgements**

The work is carried out under the Simon Associate Programme of International Centre for Theoretical Physics
(ICTP), Trieste, Italy. V.S.N, acknowledge the support received from the Aerosol Radiative Forcing over India (ARFI) project of ISRO (I-GBP), India. Authors acknowledge the Central Pollution Control Board (CPCB) for the air quality data.

**Funding:** Simon Associate Programme of International Centre for Theoretical Physics (ICTP), Trieste, Italy.

**Author contributions:** VSN conceived the research theme. VSN and FG wrote the manuscript. UKH supported the data analysis.

**Competing interests:** The authors declare that they have no conflict of interest.

**Data and materials availability:** Data are available upon request from the contact author, Vijayakumar S. Nair (vijayakumarsnair@gmail).



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
