# Peer review of "Amplification of South Asian haze by water vapour-aerosol interactions"

_Atmospheric Chemistry and Physics, 2020_

## Referee Comment (RC1) · Anonymous Referee #2 · 6 Aug 2020

This paper uses a regional climate model, RegCM4, to assess the contribution of aerosol hygroscopic growth to the total aerosol optical depth and examines various effects resulted from the aerosol-water vapor interactions in South Asia in winter. They include the positive feedbacks between aerosol hygroscopic growth-surface cooling-PBL relative humidity, the drying effects in the free troposphere, and the worsening of air quality. These effects are assessed by a set of designed model experiments targeted to isolate the change of AOD, temperature, and RH due to the radiative forcing due to the aerosol hygroscopic growth and due to the meteorological feedback. Several interesting results are presented, including the change of AOD and surface radiation due to the aerosol hygroscopic growth and resulting change of surface temperature and relative humidity. It is also interesting to see the opposite trends of RH in the PBL

and in the free troposphere that might be explained, at least partially, by the feedbacks from the interactions between aerosol and water vapor.

Having said that, there are several major concerns regarding the quality of the paper. My major and specific comments are presented below. Therefore, I recommend a major revision of the paper before it can be published on ACP.

Major comments:

1. The paper is particularly focusing on the effects of aerosol hygroscopic growth on two meteorological variables, T and RH, which is fine. But attributing the amplification of the winter haze in South Asia to the aerosol-H2O interaction is rather narrow-minded, because there is another important factor, the effects of aerosol-radiation interactions due to the presence of absorbing aerosols, which is probably a more dominant factor in amplification of the South Asian haze but was not addressed at all in the manuscript. I suggest adding one more model experiment, that is to assume all aerosols are not absorbing (e.g., single scattering albedo =1, or omit black carbon), to address that issue. 2. Feedbacks: The only met fields dealt with in this paper are surface temperature and RH. Other important met fields, such as clouds, precipitation, winds, and PBL height, are omitted. The authors argue that this is because some previous studies have explored these fields. However, at least these other effects should be mentioned and summarized in the context of comparing the magnitude of various effects on haze and air quality. 3. The title is about the amplification of South Asian haze, but there is no assessment of the haze condition in the paper, for example, the change of visibility or PM2.5 concentrations. Since the change of AOD does not equal to the change of haze conditions, it is necessary to assess the air quality in more direct way than what was presented in the paper. 4. The analysis and explanations are often confusing. It is not clear what the findings are from the present study and what are from other studies, and how consistent or different they are.

Specific comments:

Line 10-11: "This positive feedback mechanism plays an important role in the prevalence of wintertime fog and low air quality conditions over South Asia" – this point has not been demonstrated in the paper.

Line 70-74: These two sentences read like that so far, "there are very limited efforts to understand the implications of aerosol hygroscopicity on regional air quality", "in this study we use a regional climate model to assess the contribution of the aerosol hygroscopic growth to the total aerosol optical depth over the IGP". If the purpose is to fill the gaps on very limited efforts on the effects of aerosol hygroscopicity on air quality, assessing the contribution to total AOD does not help much in that regard.

Line 76-76: "we quantify the effects and feedbacks of the aerosol hygroscopic growth on the low air quality conditions over the region": This part is lacking. No quantification on such effects on air quality is specifically addressed in this paper.

Line 83-84: 50-km horizontal resolution and 18 vertical levels: This is a very coarse resolution regional model. What is the model top?

Line 89-90: Please spell out the acronyms and provide references of each scheme.

Line 113-114: The growth factor is 1.37 and 1.49 for hydrophilic BC and OC, respectively. What are the growth factors for sulfate and nitrate?

Line 120: "climate feedbacks" is used here loosely. For the limited seasonal study like this one, "meteorological feedback" is more appropriate.

Line 121-122: How the clouds are formed in RegCM without aerosol "feedbacks"? Does aerosol provide CCN for cloud formation?

Line 142: "RegCM simulates reasonably well. . ." This is a subjective statement. Howe well is reasonably well? How well is "reasonably well"? Within 20%? 50%? 100%? compared to what? Needs more quantitative description.

Line 146-147, Comparisons between RegCM simulated aerosols with measurements:

Does the simulation include the feedbacks? Figure 2 shows that the modeled aerosols are in general lower than observations taken in the earlier years. What does this mean considering the fast increase of pollutant concentrations and AOD (see intro section)? Does it mean the model underestimations would be even more severe if the comparisons are done for the same years?

Line 165: "A good agreement" – again, this is a subjective phrase. With the figure, you should be able to quantify the agreement, e.g., the model calculated RH is 5-20% higher than the observed values. Are these differences good enough for this study?

Line 171: another "reasonably well". Fig. 2f shows that the differences between simulated and observed PBLH range from 600m lower and ∼200m higher than observations. Is this reasonable?

Line 185-186: "The mean observed (ambient, dry) AOD" – how to observe dry AOD?

Line 186, sites: It would be better to follow the orders of a, b, c, d in Figure 3, i.e., Jaipur, Kanpur, Pune, and Gandhi College.

Line 193, the statement of Jaipur not showing much hygroscopic growth: The ambient AOD (0.51) doubles the dry AOD (0.26) in Jaipur (see line 186). This is quite a lot growth, certainly is not "not much".

Line 203: "matches well" – how well is well? Please give some quantitative measures, including correlation coef., bias, RMSE, for example.

Line 206-207: 30%. "since the emission fluxes of anthropogenic aerosols are seasonally invariant": this is from the model or the actual facts in the real world?

Line 207-208: "most of the observed variability of AOD is due to the variability in relative humidity": This is unjustified to attribute the variability to the RH, because you have not assessed the AOD variability due to the changes of winds, chemistry, deposition, and transport.

[Figure]

Line 213: What is the AODdry/AODambient ratio over W india? The contour lines started at 0.3 and ends at 0.6. I suggest show the full range of the ratio with the contour lines.

Line 210-228: This paragraph is somehow confusing – it mixes the results from this study with findings from other studies, and it is difficult to unravel the message. It should state what are the findings from this study and how they compared with previous studies.

Line 245: How much is the cloud cover change? This should be included in Figure 5.

Line 249: RH increases by 2-4% in eastern India. Is this insignificant? The student t-test shows that it is statistically significant. Please clarify what "significant" means.

Line 249-251, "Hence…" This statement is not evident from the paper. So far the figures and text only show the effects of aerosol hygroscopicity on RH and T, but not the pathways from land-atmosphere interactions.

Line 253-254, "In other words…" But why the effects in the eastern India and Bangladesh is much weaker than in the middle IGP where the dimming effect is the strongest (Fig 4)? Need some better explanation.

Line 261, "along with the weak entrainment of dry airmass from the free troposphere": Where is the evidence? Does RegCM show the weakening of the entrainment? What is the cause of that?

Line 261-262, "leads to an increase in relative and specific humidity at the surface": The paper only shows an increase of RH, not specific humidity.

Line 267-268, "Most reanalysis datasets have a negative bias in simulating RH during winter": The "reanalysis" datasets are based on the observation that should include all effects, unless the observations in South Asia are not input into the reanalysis and the RH is solely from the underline model simulations. Please explain.

Line 272, "drying in the 1-3 km": Fig 6 shows the RH increased by 3% at 1 km, not decreased, due to total aerosol effect.

Line 272-273, "warming of the top of the boundary layer": Where is the top of PBL? Fig 6 shows a cooling, not warming, from surface to almost 2 km due to total aerosol effect.

Line 287: "surface temperature increases": But you have been showing the temperature decreases, not increases. This is so confusing.

Line 287-288, increase in water vapour content...reported by Mukhopadhyay (2017)": Does that study use the same model with the same settings for wintertime simulation? What is your finding in this study?

Line 346-347, "present study also showed poor association with ambient AOD and PM2.5 but higher correlation is observed between the dry AOD and PM2.5": Where are the associations/correlations are shown in this paper?

Line 350, "the mass concentration is dominated by the contribution from the coarse mode aerosols": But PM2.5 is not coarse mode aerosols!

Line 306-375, "section 3.3 Implications on air quality": This entire section is very weak. There are no quantifications on the haze condition and/or PM2.5 concentration change due to the aerosol hygroscopic growth and meteorological feedbacks. Maps of change of air quality (haze and PM2.5), similar to Figure 5 for T and RH, are needed to illustrate the implications.

Line 383: The increase in PM2.5 along the IGP has not been unequivocally demonstrated.
* * *

---

## Referee Comment (RC2) · Anonymous Referee #1 · 12 Aug 2020

This paper assesses the contribution of the hygroscopic growth of aerosols to the total AOD and demonstrate that the increased surface cooling due to the hygroscopic effects of aerosols further increases the humidity in the boundary layer and thus enhances the confinement of pollutants through aerosol-boundary layer interactions. This study is a timely contribution to our growing understanding of the chemistry-weather interactions. The paper is well written and easy to follow. However, the paper lacks a comprehensive evaluation exercise necessary to allow the readers lend confidence in the scientific findings presented here. My major and minor concerns are listed below.

Major comment: Specifically, a qualitative model evaluation is not enough considering the importance of this topic. While I understand that chemical composition measurements are limited over India but the lead author is a part of the Aerosol Radiative

[Figure]

Forcing over India (ARFI) project which has done an excellent job in collecting hourly AOD and black carbon at more than 30 sites in India since early 2000s. I will suggest comparing model AOD and BC from the "Dry" and "Ambient" experiments with daily (hourly if possible) measurements of AOD and BC performed at the ARFI sites (similar to Fig. 3 for the AERONET sites). This evaluation will highlight where and when the hygroscopic properties play the most important role in the Indian context. In addition, the authors should also evaluate the model performance against available CPCB PM2.5 measurements because understanding the processes leading to poor air quality episodes is of utmost importance in this part of the world as also stated in the Introduction of this paper. I think a detailed PM2.5 evaluation during three seasons would provide a robust insight into the implications of aerosol hygroscopicity for air quality. Furthermore, diurnal profiles of surface RH and temperature should also be evaluated. It is particularly important to understand if the model is able to capture nighttime increase in RH which is one of the key factors in haze and fog formation. Therefore, I recommend a major revision before the paper can be accepted for publication in ACP.

Minor comments:

Line 11: Change low to poor.

Line 35: Change boundary layers to boundary layer.

Line 66: Does all aerosol optical properties (AOD, SSA, and asymmetry parameter) increase by a factor of 2 at RH > 80%?

Line 84: How many levels do you have in the boundary layer and are these sufficient to resolve the PBL processes?

Line 100-104: Does the model also include secondary organic aerosols?

Line 190: Are not organic carbon aerosol also emitted as hydrophobic?

Line 194-195: If aerosol hygroscopic growth is not a major factor at Jaipur, what is the reason behind large differences between Ambient and Dry AOD at Jaipur in Figure

3a.?

Line 209: I am not convinced that most of the AOD variability can be simply attributed to variability in RH. Since Ambient AOD is higher, surface temperature will be lower in Ambient experiment compared to the Dry. Consequently, PBL will be lower and lead to accumulation of aerosols in the PBL. The winds may also respond to aerosol induced thermodynamic changes and lead to different emissions of dust aerosols. All these aspects should be discussed.

Lines 249-255: It is really interesting to note that changes in temperature and RH are not located in the same place as the changes in AOD. Could you show the distribution of solar radiation reaching at the surface as well to corroborate your explanation because the largest changes in solar radiation reaching at the surface should coincide with the largest changes in AOD.

Line 314: What is the difference between radiative and climatic feedback?

Fig 8b: I think it is delta AOD-RH relationship. Please add delta symbol to the figure title. Why is deltaAOD-RH correlation highest over the Thar Desert?

Line 338-339: I think PM2.5 response will depend largely on how the PBL height changes and how aqueous-phase production of SO4 will change. Can you add some discussion on these points?

---

## Author Comment (AC1) · 7 Oct 2020

*This paper uses a regional climate model, RegCM4, to assess the contribution of aerosol hygroscopic growth to the total aerosol optical depth and examines various effects resulted from the aerosol-water vapor interactions in South Asia in winter. They include the positive feedbacks between aerosol hygroscopic growth-surface cooling-PBL relative humidity, the drying effects in the free troposphere, and the worsening of air quality. These effects are assessed by a set of designed model experiments targeted to isolate the change of AOD, temperature, and RH due to the radiative forcing due to the aerosol hygroscopic growth and due to the meteorological feedback. Several interesting results are presented, including the change of AOD and surface radiation due to the aerosol hygroscopic growth and resulting change of surface temperature and relative humidity. It is also interesting to see the opposite trends of RH in the PBL and in the free troposphere that might be explained, at least partially, by the feedbacks from the interactions between aerosol and water vapor. Having said that, there are several major concerns regarding the quality of the paper. My major and specific comments are presented below. Therefore, I recommend a major revision of the paper before it can be published on ACP.*

We thank the reviewer for the evaluation summary and the extensive comments to improve the manuscript.

**Major comments:**

*1.     The paper is particularly focusing on the effects of aerosol hygroscopic growth on two meteorological variables, T and RH, which is fine. But attributing the amplification of the winter haze in South Asia to the aerosol-H2O interaction is rather narrow-minded, because there is another important factor, the effects of aerosol-radiation interactions due to the presence of absorbing aerosols, which is probably a more dominant factor in amplification of the South Asian haze but was not addressed at all in the manuscript. I suggest adding one more model experiment, that is to assume all aerosols are not absorbing (e.g., single scattering albedo =1, or omit black carbon), to address that issue.*

We agree with the reviewer that the amplification of South Asian haze during winter is multifaceted. There are several factors, such as shallow boundary layer, weak/calm winds, high humidity and high concentration of absorbing aerosols, which contribute to this effect. Several modelling and observational studies that addressed the effect of aerosol-boundary layer interaction and direct radiative effects of composite aerosols over the IGP (Nair et al., 2007; Nair et al., 2016; Bharali et al., 2018). However, the contribution of the

hygroscopic growth of aerosols to AOD and visibility and its implications on the regional meteorology are largely unexplored.

We agree with the reviewer that aerosol-radiation interaction is the more dominant factor over this region due to the high concentration of absorbing aerosols. Experiments 1 and 3 are designed to estimate the aerosol-radiation interaction (Table 1 and Eqn. 3), which include absorbing and scattering aerosols. For that matter, we have validated the model simulated BC mass concentration (shown in Figure 2) using a network of ground-based observatories spread across the region. The total effect of aerosol-radiation interaction on meteorological parameters is discussed in Figure 5 and 6. We have explicitly mentioned in the revised manuscript that radiative effects of hygroscopic-growth of aerosols are the subset of aerosol-radiation interaction of composite aerosols (Eqn. 3 and 5).

As suggested by the reviewer, we have carried out one more simulation where we set the SSA of all the aerosols similar to that of sulphate aerosols (SSA~1). The change in temperature and relative humidity for this simulation is shown below. Compared to the ambient aerosol effect shown Figure 5c & d, it is seen that the surface temperature decreases and relative humidity increases in this idealized simulation where no absorbing aerosols are present in the atmosphere.

[Figure]

Figure: The change in 2m temperature and relative humidity for completely scattering aerosol system.

2.      *Feedbacks: The only met fields dealt with in this paper are surface temperature and RH. Other important met fields, such as clouds, precipitation, winds, and PBL height, are omitted. The*

*authors argue that this is because some previous studies have explored these fields. However, at least these other effects should be mentioned and summarized in the context of comparing the magnitude of various effects on haze and air quality.*

Complied with. We have revised Figure 5 and included the change in cloud fraction due to radiative effects of (Fig 5c) hygroscopic growth of aerosols and (Fig 5f) due to total aerosols. The discussion on change in boundary layer height and winds are included in the supplementary materials (Figure S6).

[Figure]

Figure 5: *Change in (a) 2m air temperature, (b) relative humidity and (c) cloud fraction solely due to the hygroscopic growth of aerosols (ambient-dry). Change in temperature and relative humidity (RH) due to the total effect of aerosols (dry + hygroscopic growth) is shown (c), (d) and (f). Dots indicate statistical significance of Student's t-test at a 90% level.*

[Figure]

Figure S6: *Change in (a) boundary layer height and (b) wind due to hygroscopic growth and meteorological feedback of aerosols. (c) and (d) are similar to figure (a) and (b) but for the total effect of aerosols.*

3.      *The title is about the amplification of South Asian haze, but there is no assessment of the haze condition in the paper, for example, the change of visibility or PM2.5 concentrations. Since the change of AOD does not equal to the change of haze conditions, it is necessary to assess the air quality in more direct way than what was presented in the paper.*

This is an important concern raised by the reviewer. We have revised the manuscript with the change in visibility and mass concentration of organics, carbonaceous and PM2.5 to address the impact on air quality.

We agree that AOD may not represent the haze conditions near the surface, especially when the long-range transport of aerosols occurs at high altitudes. Since, the aircraft and lidar observations over the IGP clearly showed that aerosols are mostly confined within the boundary layer during the winter season with lower contributions from the free tropospheric aerosols (Nair et al., 2016), we have used AOD and surface-level parameters like PM2.5, and visibility to investigate the implications of aerosol-induced forcing on air quality and visibility.

We have estimated the visibility using the aerosol extinction coefficient and is shown in Figure below. The low visibility conditions represent the poor air quality and haze conditions over the region. We find a 60% reduction in visibility because of the hygroscopic growth of aerosols. The frequency of occurrence of visibility shows a clear decrease for ambient aerosols, this is mostly attributed to the enhancement in aerosol extinction due to hygroscopic growth and an increase in relative humidity due to aerosol-induced surface dimming.

[Figure]

*Figure: Spatial variation of mean visibility over the Indian region for ambient feedback simulation. The percentage variation of visibility changes due to hygroscopic growth with respect to visibility due to dry aerosols is given as line contour.*

Figure below shows the frequency distribution of the daily mean visibility estimated from the aerosol extinction coefficient averaged over the Indo-Gangetic Plain. When

hygroscopic growth of aerosol is activated, the distribution peak shifts toward the lower value to a large extent, which means that there is an increase in the occurrence of the days with lower visibility.

[Figure]

Figure S7: F*requency distribution of spatially averaged visibility over Indo Gangetic Plain for Ambient$_{feedback}$ and Dry$_{feedback}$ simulations.*

As shown in the figure below, there also exists an enhancement in the PM2.5 mass concentrations due to the suppression of the boundary layer height induced by the meteorological feedbacks from the aerosol radiation interaction. This results in further degradation of the air quality of this region.

[Figure]

*Figure: Change in near-surface mass loading (μg m-3) due to the meteorological feedback of total aerosol radiative forcing for (a) carbonaceous aerosols, (b) inorganic aerosols and (c) PM2.5.*

This analysis further confirms that the air quality problems in the IGP are associated not only with the aerosol source strength but also with the ambient humidity conditions and its (aerosol) radiative forcing and meteorological feedback.

Nair, V. S., S. S. Babu, M. M. Gogoi, and K. K. Moorthy (2016), Large-scale enhancement in aerosol absorption in the lower free troposphere over continental India during spring, Geophys. Res. Lett., 43, 11,453–11,461, doi:10.1002/ 2016GL070669.

*4.     The analysis and explanations are often confusing. It is not clear what the findings are from the present study and what are from other studies, and how consistent or different they are.*
We are sorry for this ambiguity. We have found that results from the present study and earlier reports got jumbled, especially on page 9. We have revised the manuscript.

Specific comments:
*Line 10-11: "This positive feedback mechanism plays an important role in the prevalence of wintertime fog and low air quality conditions over South Asia" – this point has not been demonstrated in the paper.*
The enhancement in the AOD due to the hygroscopic growth of aerosols results in further solar dimming at the surface, reduction in surface temperature, increase in relative humidity and decrease in wind speed. The suppression of the boundary layer height due to these meteorological feedbacks and weaker winds hinders the PM2.5 dispersion resulting in the enhancement in the mass concentration and thus contributes to the degradation of the air quality over this region. This aspect is addressed in the revised manuscript and please see the reply to the major comment no: 3.

*Line 70-74: These two sentences read like that so far, "there are very limited efforts to understand the implications of aerosol hygroscopicity on regional air quality", "in this study we use a regional climate model to assess the contribution of the aerosol hygroscopic growth to the total aerosol optical depth over the IGP". If the purpose is to fill the gaps on very limited efforts on the effects of aerosol hygroscopicity on air quality, assessing the contribution to total AOD does not help much in that regard.*

The sentence is revised as "there are very limited efforts to understand the implications of aerosol hygroscopicity on regional meteorology, visibility and air quality"

We agree with the reviewer that, in the present study the effect of hygroscopic-growth of aerosols on air quality is (PM2.5) not investigated in detail. Increase in PM2.5 (Figure 9c) for aerosol forcing and meteorological feedback are relatively smaller compared to AOD. So, we have revised the section and please see the answer given for major comment 3 regarding air quality.

*Line 76-76: "we quantify the effects and feedbacks of the aerosol hygroscopic growth on the low air quality conditions over the region": This part is lacking. No quantification on such effects on air quality is specifically addressed in this paper.*

Sentence revised as "Further, we quantify the effects of the aerosol hygroscopic growth on the regional meteorology, visibility and low air quality conditions over the region."

We thank the reviewer for pointing out this drawback. In the revised manuscript, we have addressed this issue (refer the reply to major comment 3).

*Line 83-84: 50-km horizontal resolution and 18 vertical levels: This is a very coarse resolution regional model. What is the model top?*

We agree with the reviewer that this domain has a coarse spatial resolution. The model top is at 50 hPa. We have carried out higher resolution runs but they did not add much value to the problem being addressed in the paper. Considering the computation cost and memory requirements we have optimized this configuration.

*Line 89-90: Please spell out the acronyms and provide references of each scheme.*

Complied with.

"The parameterization schemes used in the simulations are (1) boundary layer: University of Washington PBL scheme, (2) Convection over land and ocean: Tiedtke scheme, (3) Radiative transfer: CCM3 (Community Climate Model) scheme, (4) cloud microphysics: Subgrid Explicit Moisture Scheme (SUBEX) and (5) Land surface: Biosphere-Atmosphere Transfer Scheme (BATS) scheme."

*Line 113-114: The growth factor is 1.37 and 1.49 for hydrophilic BC and OC, respectively. What are the growth factors for sulfate and nitrate?*

Complied with.

"at 80% relative humidity, the particle mass extinction cross-section of sulphate and nitrate increases by a factor of 3 compared to its dry value."

*Line 120: "climate feedbacks" is used here loosely. For the limited seasonal study like this one, "meteorological feedback" is more appropriate.*

We agree with the reviewer. Complied with.

*Line 121-122: How the clouds are formed in RegCM without aerosol "feedbacks"? Does aerosol provide CCN for cloud formation?*

RegCM4 doesn't have explicit aerosol-cloud interaction parameterization scheme. Clouds are formed in the model depending on the meteorological conditions and the selection of convection parameterization schemes. Actually, this indirect effect of aerosols is currently being implemented in the model.

*Line 142: "RegCM simulates reasonably well: : :" This is a subjective statement. Howe well is reasonably well? How well is "reasonably well"? Within 20%? 50%? 100%? compared to what? Needs more quantitative description.*

Sorry for these qualitative statements regarding the model validation of aerosol and meteorological parameters. That sentence was intended to mention that several earlier studies validated the meteorological parameters with direct observations over the Indian region (Usha et al., 2020; Ajai et al.,2019; Nair et al., 2012; Giorgi et al., 2012). These authors fine-tuned the model parameterization schemes and carried out a sensitivity analysis in detail.

What could be the reasonable range within which one can accept the model simulations is highly relative. For example, several GCMs simulate BC mass loading over the Indian region by almost 2 to 5 times lower than the observations and some models fail to reproduce the basic monsoon rainfall characteristics (Pan et al., 2015; Wang et al., 2018). However, we have considered the simulation as "reasonably good" if the bias is within the standard deviation of the measurements.

The sentence is revised as "Several earlier studies used RegCM to simulate the regional meteorology of South Asia (Ajay et al., 2019; Giorgi et al., 2012; Nair et al., 2012; Usha et al., 2020)."

Pan, X., Chin, M., Gautam, R., Bian, H., Kim, D., Colarco, P. R., Diehl, T. L., Takemura, T., Pozzoli, L., Tsigaridis, K., Bauer, S., and Bellouin, N.: A multi-model evaluation of aerosols over South Asia: common problems and possible causes, Atmos. Chem. Phys., 15, 5903–5928, doi:10.5194/acp-15-5903-2015, 2015.

Wang, Z., Li, G. & Yang, S. Origin of Indian summer monsoon rainfall biases in CMIP5 multimodel ensemble. *Clim Dyn* 51, 755–768 (2018). https://doi.org/10.1007/s00382-017-3953-x

*Line 146-147, Comparisons between RegCM simulated aerosols with measurements: Does the simulation include the feedbacks? Figure 2 shows that the modeled aerosols are in general lower than observations taken in the earlier years. What does this mean considering the fast increase of pollutant concentrations and AOD (see intro section)? Does it mean the model underestimations would be even more severe if the comparisons are done for the same years?*

All the comparisons shown in Figure 2 include aerosol feedback.

As far as the effect of the increasing trend in AOD and aerosol concentrations are concerned, since most of the measurements considered in this study are taken during the last 15 years, the contribution of the long-term trend in the difference between model and measurements are within the standard deviation of the data. For example, the increase in AOD during the last decade ($0.06$ decade$^{-1}$ reported by Moorthy et al., (2013)) is small compared to the mean AOD over the region (less than 10% of mean AOD over the IGP). Broadly, Sulfate is increasing over the region with a small growth rate, whereas BC at surface level is decreasing trend at ~240 ng m$^{-3}$ yr$^{-1}$. The bias due to the asynchronous model data and observations are rather small compared to the uncertainties associated with (i) emission inventories, (ii) boundary layer variation, (iii) hygroscopicity parameterization, and (iv) scavenging processes. However, compared to the earlier studies, where simulated BC mass loading was 2 to 10 times lower than the observed values, the present study simulates values within the standard deviation of the actual observations.

*Line 165: "A good agreement" – again, this is a subjective phrase. With the figure, you should be able to quantify the agreement, e.g., the model calculated RH is 5-20% higher than the observed values. Are these differences good enough for this study?*

Complied with. The sentence is revised as

"The RegCM simulated relative humidity values are approximately 10% higher than the measured RH, with a mean absolute error of 6.6% and RMSE of 7.3%. The good agreement between measured and simulated (with aerosol feedback) relative humidity over the IGP

(Fig. 2e) is important for the estimation of aerosol optical properties from simulated speciated aerosol mass concentrations."

Line 171: another "reasonably well". Fig. 2f shows that the differences between simulated and observed PBLH range from 600m lower and _200m higher than observations. Is this reasonable?

In the case of PBL intercomparison, the model bias ranges from -57 m over Bhopal to 352 m over Ranchi with a mean absolute error of 200 m (26% of mean PBLH) and RMSE of 240 m. The mean PBL height at 6 locations over the study region is 793 ± 323 m and the standard deviation of the PBL height observations varied from 100m to 670m over Guwahati and Bhopal respectively. Considering this large day to day and diurnal variations of the PBL height, the simulated PBL heights are reasonably good.

Line 185-186: "The mean observed (ambient, dry) AOD" – how to observe dry AOD?

Sorry for that mistake. We cannot "observe" the dry AOD, but we can estimate. The sentence is modified as "The mean observed AODs (simulated ambient and dry AODs) for Jaipur, Kanpur, Pune and Gandhi College are 0.53 (0.51, 0.26), 0.83 (0.81, 0.34), 0.48 (0.32, 0.15) and 1.15 (0.92, 0.36) respectively."

Line 186, sites: It would be better to follow the orders of a, b, c, d in Figure 3, i.e., Jaipur, Kanpur, Pune, and Gandhi College.

Complied with. "The mean observed AODs (simulated ambient and dry AODs) for Jaipur, Kanpur, Pune and Gandhi College are 0.53 (0.51, 0.26), 0.83 (0.81, 0.34), 0.48 (0.32, 0.15) and 1.15 (0.92, 0.36) respectively."

Line 193, the statement of Jaipur not showing much hygroscopic growth: The ambient AOD (0.51) doubles the dry AOD (0.26) in Jaipur (see line 186). This is quite a lot growth, certainly is not "not much".

We agree with the reviewer. We have modified the sentence. "Observations close to the Thar Desert (Fig. 1a, Jaipur) show relatively lower hygroscopic growth (43%) compared to other IGP stations because of the dominance of dust aerosols and relatively low humidity conditions prevailing there."

Line 203: "matches well" – how well is well? Please give some quantitative measures, including correlation coef., bias, RMSE, for example.

Complied with. The mean absolute error of 0.12 (~17% of mean AOD) and RMSE of 0.15.

*Line 206-207: 30%. "since the emission fluxes of anthropogenic aerosols are seasonally invariant": this is from the model or the actual facts in the real world?*

Since we used monthly emission data, the model emission fluxes are constant for the entire month. Though may not be the case in the real condition, we don't have high resolution (temporal and spatial) emission data over the region. The sentence is modified as "All the meteorological conditions, associated processes (winds, chemistry, deposition, and transport) and anthropogenic emissions remained the same for dry and ambient AOD simulations, except the hygroscopic growth of AOD with relative humidity for the ambient AOD."

*Line 207-208: "most of the observed variability of AOD is due to the variability in relative humidity": This is unjustified to attribute the variability to the RH, because you have not assessed the AOD variability due to the changes of winds, chemistry, deposition, and transport.*

Figure 3 clearly shows that ambient AOD has more variability (standard deviation or the coefficient of variation) than dry AOD. All the meteorological conditions, associated processes (winds, chemistry, deposition, and transport) and anthropogenic emissions remained the same for dry and ambient AOD simulations except the hygroscopic growth of AOD with RH for the latter case. Hence it is clear that relative humidity contributes to the relatively high day-to-day variability of AOD over the region.

*Line 213: What is the AODdry/AODambient ratio over W india? The contour lines started at 0.3 and ends at 0.6. I suggest show the full range of the ratio with the contour lines.*

Dry AOD contributes more than 50 to 70% of total AOD over W India. We have modified the figure and shown below.

[Figure]

*Figure 4: (a) Aerosol optical depth (AOD) simulated using RegCM4 for dry and ambient humidity conditions. Colour map indicates AOD(ambient) − AOD(dry) for without meteorological feedback and line contours show AOD(dry)/AOD(ambient) (b) Difference in aerosol direct radiative effects (DRE) at the surface for dry and ambient humidity conditions (without meteorological feedbacks).*

*Line 210-228: This paragraph is somehow confusing – it mixes the results from this study with findings from other studies, and it is difficult to unravel the message. It should state what are the findings from this study and how they compared with previous studies.*

We are really sorry for this confusion. We have revised that paragraph as

"The AOD due to hygroscopic growth ( $\Delta AOD_{RH}$) estimated using Eqn. 1 is shown in Fig. 4a. The $\Delta AOD_{RH}$ values are high over IGP, especially the eastern IGP and Bangladesh ($\Delta AOD_{RH}$) > 0.3). The ratio of AOD(Dry$_{nofeedback}$) to AOD(Ambient$_{nofeedback}$) shown as contours in Figure 4a indicates that dry aerosols contribute 50 to 60% of the AOD over most of the Indian sub-continent (Fig. 4a), except over western India, where the contribution of hydrophobic dust to the total AOD is high even during winter. The contribution of the aerosol hygroscopic growth is more than 50% of the total AOD over the northern Indian Ocean, which is attributed to the dominance of hydrophilic aerosols (sea salt) and high relative humidity. It is interesting to note that the increase in AOD solely due to hygroscopic growth ($\Delta AOD_{RH}$) = AOD(ambient) − AOD(dry)) is ~0.23 over the IGP (Fig. 4a), which is higher than the global mean AOD (0.155 ± 0.018) reported by Watson-Parris et al., (2020) using various satellite observations. Mandariya et al., (2020) have reported

that wintertime aerosols over central IGP show an increase in particle diameter by 1.34±0.07 times as that of the dry particle at 85% relative humidity. This increase in particle diameter corresponds to a doubling of the scattering coefficient for anthropogenic aerosols (McInnes et al., 1998). In the present study the mean dry AOD increases almost 1.7 times due to relative humidity over IGP, which is slightly lower than the estimated AOD for the hygroscopic factor reported by Mandariya et al., (2020). Since organic aerosols dominate the aerosol system over IGP (Ram et al., 2012), there could be a substantial decrease in the hygroscopic growth of particles as discussed by several studies. Mandariya et al., (2020) have reported that 150 nm particles are more hygroscopic than 100 nm particles due to the higher organic fraction present in the 100 nm particle mass concentrations. In addition, although most of the observational studies consider 40% as the reference humidity level (dry) for estimating the hygroscopic growth, the model estimates the dry AOD for zero relative humidity. Titos et al., (2016) reported that the assumption of no growth at 40% humidity may contribute higher uncertainty in the estimates of measurement-based hygroscopic growth."

*Line 245: How much is the cloud cover change? This should be included in Figure 5.*

Complied with. Figure 5 is revised as shown below. "The cloud cover increases by 3 to 5 % over IGP due to ambient aerosol forcing (Figure 5f)."

[Figure]

*Figure 5: Change in (a) 2m air temperature, (b) relative humidity and (c) cloud fraction solely due to the hygroscopic growth of aerosols (ambient-dry). Change in temperature and relative humidity (RH) due to the total effect of aerosols (dry + hygroscopic growth) is shown (c), (d) and (f). Dots indicate statistical significance of Student's t-test at a 90% level.*

*Line 249: RH increases by 2-4% in eastern India. Is this insignificant? The student t-test shows that it is statistically significant. Please clarify what "significant" means.*

We have mentioned inline 249 that "Although AOD and $\Delta AOD_{RH}$ are high, the change in relative humidity due to hygroscopic growth of aerosols is negligible over Bangladesh and Eastern IGP." This inference is made based on Figure 5b, which shows <1% change in RH and it is statistically insignificant as shown by Student t-test. However, for the total aerosol effect (Figure 5d) there is a 2-4% increase in RH and it is statistically significant also (as mentioned by the reviewer). We assume that the reviewer considered Figure 5d (new figure 5e) instead of figure 5b. To avoid this ambiguity, we have revised the sentence as

"Although AOD and $\Delta AOD_{RH}$ are high, the change in relative humidity due to hygroscopic growth of aerosols (Figure 5b) is negligible over Bangladesh and Eastern IGP."

*Line 249-251, "Hence: : :" This statement is not evident from the paper. So far the figures and text only show the effects of aerosol hygroscopicity on RH and T, but not the pathways from land-atmosphere interactions.*

The aerosol induced decrease in surface-reaching solar radiation and its influence on sensible and latent flues (Bowen ratio) is already discussed in Bharali et al., (2019), hence not repeated here. Our results also showed a similar change in heat fluxes over IGP during winter. We have modified the sentence as "Bharali et al., (2019) have reported that the strength of land-atmosphere interactions through the exchange of heat and moisture fluxes plays a major role in deciding the aerosol induced dimming effects on meteorology (more details are available in Li et al., 2017)"

*Line 253-254, "In other words: : :" But why the effects in the eastern India and Bangladesh is much weaker than in the middle IGP where the dimming effect is the strongest (Fig 4)? Need some better explanation.*

We thank the reviewer for this very important point which we missed in the initial analysis. It is interesting to note that aerosol loading and clear-sky radiative forcing showed very high values over the eastern IGP. In contrast, the change in surface temperature and relative humidity due to aerosol induced surface dimming was high over central IGP with a moderate change over eastern IGP. To further understand this, the change in solar radiation reaching at the surface for with and without aerosol conditions is shown in the figure below. Though the AOD and aerosol direct radiative forcing are high values over the eastern IGP, the presence of relatively high cloud fraction over the eastern IGP mask the aerosol-induced surface cooling. So, the change in surface temperature did not show a similar pattern as that of AOD. Whereas the change in shortwave flux at the surface (due to clouds and aerosols together) showed an almost similar pattern as that of change in temperature and humidity.

[Figure]

*Figure S4: (a) Spatial variation of change in surface-reaching solar radiation for with and without aerosol conditions. (b) variation of AOD (color) and cloud fraction (line contour) for aerosol feedback simulation.*

*Line 261, "along with the weak entrainment of dry airmass from the free troposphere": Where is the evidence? Does RegCM show the weakening of the entrainment? What is the cause of that?*

The figure below shows the change in the model vertical wind due to aerosol effects. The large-scale entrainment decreases above 2 km due to aerosol feedback. The convection also weakens due to the decreased surface temperature following the solar dimming which reduces the energy required for the upward motion the air parcel.

[Figure]

*Figure S5: Vertical profiles of change in specific humidity due to the total aerosol effect (Ambient$_{feedback}$ -Ambient$_{nofeedback}$) (black line) and only due to hygroscopic growth of aerosols (Ambient$_{feedback}$ - dry$_{feedback}$) (red line).*

*Line 261-262, "leads to an increase in relative and specific humidity at the surface": The paper only shows an increase of RH, not specific humidity.*

Sorry for taking it granted. We have included this figure in the new supplementary file (Figure S5).

[Figure]

*Line 267-268, "Most reanalysis datasets have a negative bias in simulating RH during winter": The "reanalysis" datasets are based on the observation that should include all effects, unless the observations in South Asia are not input into the reanalysis and the RH is solely from the underline model simulations. Please explain.*

As we mentioned in the manuscript, Chatani and Sharma, (2018) have clearly shown that NCEP and ECMWF underestimate relative humidity over the IGP. Figure 3 of Chatani and Sharma, (2018) is shown below for the reviewer's reference. It indicates that RH measurements from this region are not being accepted by the stringent data assimilation criteria.

[Figure]

The figure is taken from Chatani and Sharma, (2018).

*Line 272, "drying in the 1-3 km": Fig 6 shows the RH increased by 3% at 1 km, not decreased, due to total aerosol effect.*

We agree. Revised as "drying in the 1.5 to 3.5 km"

*Line 272-273, "warming of the top of the boundary layer": Where is the top of PBL? Fig 6 shows a cooling, not warming, from surface to almost 2 km due to total aerosol effect.*

Revised the sentence.

"In contrast to boundary layer moistening, drying in the lower free troposphere (1.5 to 3.5 km) by 1 to 2% was attributed to the aerosol induced warming, which decreases the humidity and suppresses the mixing of airmass between the free troposphere and boundary layer."

*Line 287: "surface temperature increases": But you have been showing the temperature decreases, not increases. This is so confusing.*

Sorry for that mistake. Corrected.

*Line 287-288, increase in water vapour content: : :reported by Mukhopadhyay (2017)": Does that study use the same model with the same settings for wintertime simulation? What is your finding in this study?*

Mukhopadhyay et al., (2017) used in-situ measurements of humidity over the Indian region from a network of observatories spread across the country to study the trend in relative and specific humidity for all the seasons. Our model simulations suggest that the aerosol-radiation interaction and associated meteorological feedback could be partially contributing to the wetting of boundary layer (a positive trend in RH) and drying of the lower free troposphere (a negative trend in RH) reported by the Mukhopadhyay et al., (2017).

*Line 346-347, "present study also showed poor association with ambient AOD and PM2.5 but higher correlation is observed between the dry AOD and PM2.5": Where are the associations/correlations are shown in this paper?*

Sorry for that omission. Figure below shows the correlation between the wet and dry AOD with PM2.5. Please note that PM2.5 is not sensitive to humidity change or PM2.5 is generally estimated as dry mass loading of particle below 2.5 µm size. So, it is expected that dry AOD has a better association with PM2.5 than ambient AOD. This result assumes importance, for the estimation of PM2.5 from the satellite observations or to find an empirical relation to covert AOD to PM2.5 for air quality applications. It is clear from this figure that high relative humidity deteriorates the association between AOD and PM2.5, which is a very significant contributor over the IGP. The accurate assessment of hygroscopic growth factor is essential to retrieve PM2.5 over the IGP using satellite sensors.

[Figure]

*Line 350, "the mass concentration is dominated by the contribution from the coarse mode aerosols": But PM2.5 is not coarse mode aerosols!*

We mentioned there that, generally optical properties of aerosols are more sensitive to the fine mode aerosols, whereas mass concentration is more contributed by coarse mode aerosols. We have removed that general sentence from the manuscript.

*Line 306-375, "section 3.3 Implications on air quality": This entire section is very weak. There are no quantifications on the haze condition and/or PM2.5 concentration change due to the aerosol hygroscopic growth and meteorological feedbacks. Maps of change of air quality (haze and PM2.5), similar to Figure 5 for T and RH, are needed to illustrate the implications.*

We agree with the reviewer. This aspect is discussed in the reply to major comment 3.

*Line 383: The increase in PM2.5 along the IGP has not been unequivocally demonstrated.*

We have removed "and PM2.5" from that sentence.

---

## Author Comment (AC2) · 7 Oct 2020

Referee #1

*This paper assesses the contribution of the hygroscopic growth of aerosols to the total AOD and demonstrate that the increased surface cooling due to the hygroscopic effects of aerosols further increases the humidity in the boundary layer and thus enhances the confinement of pollutants through aerosol-boundary layer interactions. This study is a timely contribution to our growing understanding of the chemistry-weather interactions. The paper is well written and easy to follow. However, the paper lacks a comprehensive evaluation exercise necessary to allow the readers lend confidence in the scientific findings presented here. My major and minor concerns are listed below.*

We thank the reviewer for the encouraging comments.

*Major comment: Specifically, a qualitative model evaluation is not enough considering the importance of this topic. While I understand that chemical composition measurements are limited over India but the lead author is a part of the Aerosol Radiative Forcing over India (ARFI) project which has done an excellent job in collecting hourly AOD and black carbon at more than 30 sites in India since early 2000s. I will suggest comparing model AOD and BC from the "Dry" and "Ambient" experiments with daily (hourly if possible) measurements of AOD and BC performed at the ARFI sites (similar to Fig. 3 for the AERONET sites). This evaluation will highlight where and when the hygroscopic properties play the most important role in the Indian context.*

We are sorry for being qualitative in model validation. We have included the mean absolute bias and RMSE of the modelled parameters compared to direct observation. As suggested by the reviewer, RegCM4 simulated AOD and BC are compared with the ARFINET measurements from December 2016 to February 2016. Modelled ambient AOD values are close to the observed values over all the station as shown in the below Figure. The measurement-based estimation of dry AOD is still challenging over this region because of the lack of information on the vertically resolved chemical composition and/or hygroscopicity of aerosols. BC mass concentration over Agra, Jaisalmer, Udaipur and Hyderabad also showed good association with observed values. This comparison clearly shows that the model is able to simulate the aerosol loading over the Indian region. As reviewer correctly pointed out, a synergy of more dedicated experiments and modelling is essential to delineate the various pathways of aerosol-climate interaction over the region.

[Figure]

Figure S2: *Variation of aerosol optical depth measured using multi-wavelength radiometer installed as a part of Aerosol Radiative Forcing over India Project (ARFINET) and modeled using RegCM4 from December 2015 to February 2016.*

[Figure]

Figure S3: *Temporal variation of BC mass loading measured at 4 ARFINET stations and modeled using RegCM4 over Agra, Jaisalmer, Udaipur and Hyderabad from December 2015 to February 2016.*

*In addition, the authors should also evaluate the model performance against available CPCB PM2.5 measurements because understanding the processes leading to poor air quality episodes is of utmost importance in this part of the world as also stated in the Introduction of this paper. I think a detailed PM2.5 evaluation during three seasons would provide a robust insight into the implications of aerosol hygroscopicity for air quality.*

As reviewer suggested, PM2.5 simulated using RegCM is validated using the CPCB measurements made at 19 urban centres over the Indian region. In general, the model underestimates PM2.5 values, especially at Delhi, Muzaffarpur, Gaya, Thiruvananthapuram, Lucknow, Varanasi, Agra and Guwahati. Since CPCB measurements are carried out at the urban hotspots and close to the city pollution, models simulations at coarse spatial resolution (50 km) may not simulate the magnitude of city pollution accurately.

[Figure]

Figure S1: *Inter-comparison of PM2.5 simulated using RegCM4 with measurements from 19 stations maintained by Central Pollution Control Board (CPCB) India (www.cpcb.nic.in).*

*Furthermore, diurnal profiles of surface RH and temperature should also be evaluated. It is particularly important to understand if the model is able to capture nighttime increase in RH which is one of the key factors in haze and fog formation.*

As suggested by the reviewer the diurnal variation of the temperature and relative humidity over the 4 stations in northern India is given below. As shown in the figure, the modelled RH shows a positive bias and temperature shows a negative bias over all the stations (Agra, Delhi, Patna and Lucknow). Though there exists a positive bias in the

modelled RH, RegCM4 could capture the diurnal variation with daytime low and nighttime high over the region. As Reviewer correctly pointed out the early morning high in the RH is one of the key factors for the haze and fog conditions prevailing over the IGP.

[Figure]

*Figure: Diurnal variation of measured (black) and modelled (red) relative humidity at (a) Agra, (b) Delhi, (c) Patna and (d) Lucknow.*

[Figure]

*Figure: Diurnal variation of measured (black) and modelled (red) 2m temperature at (a) Agra, (b) Delhi, (c) Patna and (d) Lucknow.*

*Therefore, I recommend a major revision before the paper can be accepted for publication in ACP.*

We thank the reviewer for the encouraging comments.

Minor comments:

*Line 11: Change low to poor.*

Complied with.

*Line 35: Change boundary layers to boundary layer.*

Complied with.

*Line 66: Does all aerosol optical properties (AOD, SSA, and asymmetry parameter) increase by a factor of 2 at RH > 80%?*

Sorry for that mistake. AOD shows a 2-fold increase, whereas SSA and asymmetry parameters show relatively weak dependence on RH.

Text revised as "The optical properties of aerosol (AOD/extinction coefficients) is enhanced by more than 2 times at higher relative humidity (>80%) conditions, which has a strong dependence on the relative dominance of organic and inorganic species (chemical composition) and size of the particle."

*Line 84: How many levels do you have in the boundary layer and are these sufficient to resolve the PBL processes?*

RegCM4 has 5 levels below 1km and model cannot resolve the PBL processes explicitly.

University of Washington (UW) PBL scheme is used in RegCM4 for the representation of the PBL processes. The layer below which the buoyancy flux cannot be more negative than -0.5 of the layer-mean buoyancy fluxes is estimated as the PBL top height. Which means that the vertical profile of the virtual temperature in the lower atmosphere stabilizes in such a way that at the boundary layer top the buoyancy flux is opposite and half of the layer mean buoyancy flux (Elguindi et al., 2014).

*Line 100-104: Does the model also include secondary organic aerosols?*

RegCM4 doesn't have an explicit SOA scheme, rather OC is multiplied with 1.25 to account for secondary organic carbon.

*Line 190: Are not organic carbon aerosol also emitted as hydrophobic?*

The OC is emitted as hydrophobic and then it transformed to hydrophilic at an ageing time of 1.15 days.

*Line 194-195: If aerosol hygroscopic growth is not a major factor at Jaipur, what is the reason behind large differences between Ambient and Dry AOD at Jaipur in Figure*

Sorry for the ambiguity. We did not rule out the RH effect over Jaipur, where hygroscopic growth was relatively smaller compared to the other IGP locations. Aerosols over Jaipur are not as hygroscopic as the central and eastern IGP aerosols.

We have modified the sentence. "Observations close to the Thar Desert (Fig. 1a, Jaipur) depict relatively lower hygroscopic growth (43%) compared to other IGP stations because of the dominance of dust aerosols and relatively low humidity conditions prevailing there."

*Line 209: I am not convinced that most of the AOD variability can be simply attributed to variability in RH. Since Ambient AOD is higher, surface temperature will be lower in Ambient experiment compared to the Dry. Consequently, PBL will be lower and lead to accumulation of aerosols in the PBL. The winds may also respond to aerosol induced thermodynamic changes and lead to different emissions of dust aerosols. All these aspects should be discussed.*

Sorry for this ambiguity.

We have considered ambient and dry simulations without climate feedback (exp 1 & 2) to estimate the coefficient of variation of $AOD_{dry}$ and $AOD_{ambient}$. Since the climate feedback is switched off, meteorology (PBL, wind, precipitation and RH) is invariant or not affected by the aerosol forcing. All the meteorological conditions, associated processes (winds, chemistry, deposition, and transport) and anthropogenic emissions remained the same for dry and ambient AOD simulations except the hygroscopic growth of AOD with RH for the latter case. Hence it is clear that relative humidity contributes to the relatively high day-to-day variability of AOD over the region.

*Lines 249-255: It is really interesting to note that changes in temperature and RH are not located in the same place as the changes in AOD. Could you show the distribution of solar radiation reaching at the surface as well to corroborate your explanation because the largest changes in solar radiation reaching at the surface should coincide with the largest changes in AOD.*

Complied with. It is interesting to that aerosol loading and clear-sky radiative forcing showed very high values over the eastern IGP. In contrast, the change in surface temperature and relative humidity due to aerosol induced surface dimming was high over central IGP with a moderate change over eastern IGP. To further understand this, the

change in solar radiation reaching at the surface for with and without aerosol conditions is shown in the figure below. Though the AOD and aerosol direct radiative forcing (clear sky) are high values over the eastern IGP, the presence of relatively high cloud fraction over the eastern IGP mask the aerosol-induced surface cooling. So, the change in surface temperature did not show a similar pattern as that of AOD. Whereas the change in shortwave flux at the surface (due to clouds and aerosols together) showed an almost similar pattern as that of change in temperature and humidity.

[Figure]

Figure: The spatial variation of (a) AOD, (b) cloud fraction and (c) net shortwave flux at surface.

*Line 314: What is the difference between radiative and climatic feedback?*

Replaced "radiative and climatic feedback" with "meteorological feedback"

*Fig 8b: I think it is delta AOD-RH relationship. Please add delta symbol to the figure title. Why is deltaAOD-RH correlation highest over the Thar Desert?*

We thank the reviewer for pointing out this mistake. Figure caption is corrected now.

*Line 338-339: I think PM2.5 response will depend largely on how the PBL height changes and how aqueous-phase production of SO4 will change. Can you add some discussion on these points?*

As reviewer correctly pointed out, the variation in PM2.5 is mostly associated with the change in PBL height change due to aerosol-induced surface cooling and other meteorological feedback. We have included the variation of carbonaceous, organics and PM2.5 aerosol mass concentration due to aerosol forcing in the below figure. The aerosol

concentration increases along the IGP due to aerosols due to aerosol induced weak ventilation over the region. This further confirms that the air quality problems of IGP have not only associated with aerosol source strength but also to its (aerosol) radiative forcing and meteorological feedback.

[Figure]

*Figure: Change in near surface mass loading (μg m⁻³) due to the meteorological feedback of total aerosol radiative forcing for (a) carbonaceous aerosols, (b) inorganic aerosols and (c) PM2.5.*